# Fabrication of High-Acyl Gellan-Gum-Stabilized β-Carotene Emulsion: Physicochemical Properties and In Vitro Digestion Simulation

**DOI:** 10.3390/foods11121742

**Published:** 2022-06-14

**Authors:** Yuecheng Meng, Linyue Hang, Sheng Fang, Yanhua Li, Xuejiao Xu, Fan Zhang, Jie Chen

**Affiliations:** 1School of Food Science and Biotechnology, Zhejiang Gongshang University, Hangzhou 310018, China; mengyc@zjgsu.edu.cn (Y.M.); Antheaspirktimothy@163.com (L.H.); fangsheng@zjgsu.edu.cn (S.F.); liyanhua@zjgsu.edu.cn (Y.L.); zf809214821@163.com (F.Z.); 2College of Biology and Environmental Engineering, Zhejiang Shuren University, Hangzhou 310015, China; xuxuejiao@zjsru.edu.cn

**Keywords:** high-acyl gellan gum, emulsion, β-carotene, in vitro simulated digestion

## Abstract

The β-carotene emulsion system using high-acyl gellan gum (HA) as an emulsifier was fabricated and systematically studied. The stability and stabilizing mechanism of the emulsion using medium-chain triglyceride as oil phase with a water-oil mass ratio of 9:1 under different physicochemical conditions of heat, pH, and ions were investigated by analyzing mean particle size (MPS), emulsion yield (EY), and dynamic stability. The effects of the HA-β-carotene emulsion system on the bioaccessibility of β-carotene in vitro were conducted. During the simulated oral digestion stage (SODP) and simulated gastric digestion stage (SGDP), the emulsion systems stabilized with different HA contents showed good stability, and the changes of MPS and zeta potential (ZP) were within 2.5 μm and 3.0 mV, respectively. After entering the simulated intestinal digestion phase (SIDP), β-carotene was released from oil droplets and formed micelles with bile salts, phospholipids, etc. HA-β-carotene emulsion can enhance the release rate of free fatty acid (FFA), which ultimately affects the β-carotene bioaccessibility. These results indicate that HA can be used to prepare carotene emulsion and improve its bioavailability. The study provides a reference for the application of HA as a natural emulsifier and the delivery of β-carotene.

## 1. Introduction

β-carotene, a precursor to the synthesis of vitamin A, has several health benefits, such as anticancer and antioxidant properties and cardiovascular disease prevention [1,2]. However, β-carotene is difficult to dissolve in water and is sensitive to light, heat, and oxygen, which results in its low bioaccessibility [3]. These characteristics greatly limit its application in the food and nutrition industries [4,5].

In recent years, many methods including delivery systems based on oil-in-water emulsions [6,7] have been developed to improve the solubility, stability, and bioavailability of β-carotene. Different emulsifier systems, such as soy protein isolate and *Pleurotus eryngii* polysaccharide [8], lactoferrin and alginate [9], soybean lecithin and phospholipids [10], and chitosan–chlorogenic acid complexes [11], have been successfully used to prepare β-carotene functional emulsions. The development of novel natural emulsifiers especially polysaccharides to improve the bioavailability of β-carotene is crucial to its application in the food and nutrition industry. Different emulsifiers might lead to different digestion behaviors [6,12], and for this reason, there is an interest in understanding how the emulsion system affects the digestion and absorption characteristics of β-carotene. 

Gellan gum is a linear anionic microbial heteropolysaccharide produced by fermentation of *Sphingomonas paucimobilis* isolated from the plant tissue of *Elodea nuttallii* under aerobic conditions. It can be divided into high-acyl gellan gum (HA) and low-acyl gellan gum (LA) according to its acyl content [13,14]. Although the application of gellan gum is mainly focused on its gel properties, studies have also shown that it has good emulsifying properties and can be used as a natural emulsifier in the food industry [15,16]. Compared with synthetic emulsifiers, gellan gum is safe, PEG-free, and biodegradable. However, the use of gellan gum as a natural emulsifier to prepare β-carotene emulsions and, most interesting, their digestion behaviors has not been studied.

Therefore, this study aims to determine the feasibility of HA gellan gum to prepare β-carotene emulsions and to investigate the digest behavior of the emulsion in vitro. The physicochemical properties of the emulsion were characterized and discussed. Then, the corresponding in vitro digestion model was established to study the digestive properties of the β-carotene emulsion. The study will provide a reference for the application of gellan gum as a natural emulsifier and the digestion and absorption of β-carotene.

## 2. Materials and Methods

### 2.1. Materials 

A commercial high acyl gellan gum, Kelcogel LT100 (CPKelco, Atlanta, GA, USA; Lot# 9L0058A CAS No:7101-52-1) was used in this study as well as medium-chain triglycerides (MCT, food-grade, Shanghai Yuanye Biotechnology Co., Ltd., Shanghai, China), derived from palm kernel and containing fatty acids that have a chain length of 8–12 carbon atoms; β-carotene (purity > 96%, food-grade, Zhejiang Xinchang Pharmaceutical Co., Ltd., Hangzhou, China); n-hexane, chloroform, sodium chloride, sodium hydroxide, hydrochloric acid, ethanol, potassium chloride, sodium dihydrogen phosphate, sodium sulfate, calcium chloride (analytical purity, Sinopharm Group Chemical Reagent Co., Ltd., Shanghai, China), α-amylase, pepsin, trypsin, glucosidase, bile salt, and Nile red dye (Sigma-Aldrich, Inc., St. Louis, MO, USA).

### 2.2. Preparation of β-Carotene Emulsion

The HA powder was directly mixed with pure water to prepare aqueous solution with different concentration gradients (final emulsion concentration gradients were 0.050%, 0.100%, 0.150%, 0.175%, 0.200% *v*/*w*). The solution was heated and stirred using a magnetic stirrer for 2 h and was equilibrated overnight before being used as the water phase. 

We next dissolved a certain amount of β-carotene (the mass concentration of β-carotene is 0.1%) in MCT oil to obtain the oil phase. The mixture was stirred in the absence of light until it was completely dissolved. The water phase containing HA was then mixed with the oil phase in a ratio of 9:1 (*w*/*w*). The final emulsion was prepared using a high-speed shear machine (T25, Fluko, Shanghai, China) at 18,000 rpm for 8 min followed by ultrasonic emulsification (Scientz, Ningbo, China) at 450 W for 8 min and with a 5 s pause time at room temperature. 

### 2.3. Mean Particle Size and Emulsion Yield

The mean particle size (MPS) and distribution of the emulsion were measured by a laser particle analyzer (Malvern 2000, Malvern Instruments Ltd., Worcestershire, UK). The surface mean diameter (d3,2) was selected.

The emulsion yield was determined according to the method of Teng et al. [17] with mirror modifications. Then, 3.0 mL n-hexane was added into 1.0 mL emulsion and was mixed under vortex for 30 s. Then, the mixture was centrifuged for 5 min under 10,000× *g* (H1850R, Xiangyi, Changsha, China). The supernatant was collected in a 10 mL volumetric flask, and the volume was filled with n-hexane. The absorption value of β-carotene was measured at 450 nm using a UV spectrometer (UV-2600, Shimadzu Co., Kyoto, Japan). The content of unwrapped β-carotene in the supernatant was calculated according to a standard curve. The emulsion yield (*EY*) was then calculated using the following equation
(1)$$EY(\%)=\frac{\beta -\mathrm{carotene}\;\mathrm{content}\;\mathrm{in}\;\mathrm{emulsion}}{\mathrm{Total}\;\beta -\mathrm{carotene}\;\mathrm{content}}\times 100$$

### 2.4. Viscosity Characteristics 

Rheological measurements were performed using an MCR302 rotational rheometer (MCR302, Anton Paar, Ostfildern, Germany). A double-cone measuring geometry (radius 34.14 mm, biconical angle 10°) was selected, and the flow curves were obtained by the ascending step procedure, where the shear rate varied between 0 and 300 s^−1^ at 25 °C [18]. 

### 2.5. Turbiscan Stability Test

The emulsions were monitored by a Turbiscan Lab-Expert analyzer (Turbiscan Lab-Expert, Formulaction, Toulouse, France). The HA-β-carotene emulsion with different concentrations of HA was placed in a Turbiscan special long-barrel scanning sample cell with a height of 40 mm. The scanning temperature was set at 45 ± 0.5 °C, and the scanning time was 6 h with a 1 h interval.

### 2.6. Physical Stability Test against Environmental Changes

The stability against heat, ions, and pH of the β-carotene emulsion (0.175%, *w*/*v*) was tested according to a previous study with some modifications [7]. For heat stability, the sample was placed into a water bath for 1 h at 35, 60, and 85 °C, respectively. Then, a 5 mL sample was taken out and cooled down immediately for characterization every 10 min. For ions stability, a certain amount of NaCl (0, 0.05%, 0.10%, 0.15%, and 0.20% *v*/*v*) was added into the emulsion to reach certain ion concentration. The MPS and zeta-potential (ZP) were measured as above (the emulsion was diluted by 200 times using the same concentration of NaCl). For pH stability, the pH of the HA-β-carotene emulsion (0.175%, *w*/*v*) was adjusted to 2, 3, 4, 5, 6, and 7 with 0.1 mol/L HCl or 0.1 mol/L NaOH solution. The mixture was equilibrated at room temperature for 1 h. The MPS and ZP were measured accordingly (the emulsion was diluted by 200 times using the same pH solution). 

### 2.7. In Vitro Digestion Test

According to the conditions listed in Appendix A, digestive juice of three stages, including the simulated oral digestion phase (SODP), the simulated gastric digestion phase (SGDP), and the simulated intestinal digestion phase (SIDP), were prepared. The 5 mL HA-β-carotene emulsion was mixed with 5 mL simulated oral digestive juice. The pH value was adjusted to 7.0 and stirred at 100 rpm for 3 min. Then, 10 mL of simulated gastric juice was added to the digestive juice after SODP, and the mixture was adjusted with concentrated hydrochloric acid to pH 2.0. The SGDP was maintained for 1 h under 100 rpm stirring. After SGDP, the pH of the system was adjusted to 7.0. Finally, 15 mL of simulated intestinal juice was prepared and added to simulate the SIDP. We continuously added 0.2 mol/L NaOH solution to keep the pH value of the digestive system at 7.0 within 2 h of digestion. The amount of NaOH solution consumed was recorded at different times during the experiment. The degree of fat droplets hydrolysis to free fatty acids (FFA) in the presence of pancreatic lipase was calculated by the consumption of NaOH in different time intervals (2, 5, 10, 20, 30, 60, and 120 min). Throughout the digestion process, the temperature remains at 37 °C.

### 2.8. Micromorphology of Emulsion

A laser confocal microscope (LSM710, Carl Zeiss AG, Oberkochen, Germany) was applied to observe the morphology of droplets in the emulsion during digestion. A dyeing solution of Nile red in ethanol (0.01%, *w*/*w*) was prepared [19]. The emulsions were sampled and dyed. The stained sample was placed on the slide and covered with the cover glass on the inverted microscope. The excitation wavelength and the absorption wavelength of the laser confocal microscope were set at 488 and 549–651 nm, respectively.

### 2.9. Bioaccessibility Determination

The bioaccessibility of β-carotene was measured according to the method of Qian et al. [20] with minor modifications. The sample was digested in SIDP and then centrifuged (H1850R, Cence Co., Ltd., Changsha, China) for 40 min at 25 °C and 10,000× *g*. After centrifugation, the sample was separated into a precipitate phase at the bottom and a clear micellar phase in the middle. The middle phase was assumed to consist of mixed micelles that solubilized the bioactive component β-carotene. Aliquots were collected directly from raw digesta or the middle phase of centrifuged samples. The sample was vortexed with 8 mL n-hexane and centrifuged at 25 °C for 10 min at 3500 r/min. The *n*-hexane phase after centrifugation was then analyzed using a UV—visible spectrophotometer at 450 nm. A cuvette containing pure *n*-hexane was used as a reference cell to zero the spectrophotometer. The concentration of β-carotene in the emulsion is calculated from a previously prepared calibration curve. The bioaccessibility (%) was calculated using the following equation: (2)$$\mathrm{Bioaccessibility}\;(\%)=\frac{C_{\mathrm{middle}\;\mathrm{phase}}}{C_{\mathrm{raw}\;\mathrm{digesta}}}\times 100\%$$

### 2.10. Statistical Analysis 

All experiments were repeated three times, and the values were described as the mean ± SD. Data were analyzed using SPSS 19.0 software (IBM, Armonk, NY, USA) for F test and analysis of variance using the Duncan analysis method (*p* < 0.05).

## 3. Results and Discussion

### 3.1. Effects of HA Contents on Properties of the β-Carotene Emulsion

The effects of different HA concentrations on the MPS and EY are shown in Figure 1A. When the concentration of HA increased from 0.05% to 0.10%, the MPS of the emulsion decreased sharply from 7.2 ± 0.2 µm to 4.5 ± 0.2 µm. With a further increase in the HA concentration, the MPS decreased with slow trends. With the increase of HA, the EY gradually increased. When the HA concentration reached 0.175%, the MPS and EY of the emulsion reached the lowest value of 4.0 ± 0.8 µm and the highest value of 84.4 ± 1.6%, respectively. In the gum-arabic-stabilized emulsion prepared by Ozturk et al., the MPD was also about 3 µm when the gum arabic concentration was in the range of 0–0.20% [21]. The photographs of the emulsion stored for different days (0–20 d) are shown in Figure 1B. When the HA concentration is larger than 0.100%, the emulsions show good stability without visible settling or creaming during 20 days of storage at room temperature. The MPS of all emulsions increased with the increase in storage days, but they were all in the stable range. These results indicated that HA can be used as an emulsifier to prepare a β-carotene emulsion. These emulsions showed an increase in absolute zeta potential values as gellan gum content increased, which indicated that electrostatic forces contributed to emulsion stability since droplets intensively charged can repel each other, maintaining the stability of the system [22,23]. Vilela et al. also found electrostatic forces contributed to gellan emulsion stability [24]. Moreover, hydrophilic gellan gum could be adsorbed on the interface of oil drop, which may inhibit the accumulation of lipid droplets and enhance the stability of the emulsion, similar to that of gum arabic [25]. Overall, when the concentration of HA is 0.175%, a stable β-carotene emulsion can be prepared.

### 3.2. Effect of HA Concentration on the Viscosity of HA-β-Carotene Emulsion

The relationship between the shear stress and the rate of different HA concentrations is shown in Figure 2A. As the shear rate increases, the shear stress gradually increases. When HA concentration is low, the system performed close to a Newton fluid. With the increase in HA concentration, the pseudoplastic property of the emulsion was enhanced. The emulsion showed shear-thinning characterization. When the shear rate was 100 s^−1^, the apparent viscosity of the emulsion system increased rapidly from 75 Pa·s to 850 Pa·s with the increase in HA concentration. When the concentration of polysaccharides is higher than the critical viscosity concentration and lower than the critical flocculation concentration, the movement of the oil droplets is blocked, which increases the stability of the emulsion [26]. Many researchers have found that viscosity plays an important role in stabilizing emulsion systems, such as in emulsions stabilized by gum arabic and xanthan gum [27,28,29]. The high viscosity of HA-β-carotene emulsion is beneficial to the stability of the emulsion.

### 3.3. Effect of HA Concentration on the Stability of HA-β-Carotene Emulsion

The influence of HA concentration on the dynamic stability of the emulsion is shown in Figure 3. The multiple-light-scattering method (Turbiscan Lab) was always used to test the stability of emulsion without damaging the sample [30,31]. In this method, the changes in back-scattering light (Δ*BS*) were recorded to evaluate the small difference in opacity [31]. As the concentration of HA changes, the emulsion exhibits different Δ*BS* characteristics during the scan time. When the concentration of HA is between 0.100% and 0.150%, the changes in Δ*BS* value show that the emulsion has an obvious creaming phenomenon. When the HA concentration is 0.175%, there are almost no changes in the Δ*BS* value. However, when the HA concentration is further increased to 0.200%, the change of Δ*BS* value becomes larger than the HA concentration of 0.175%. The smaller the change in the Δ*BS* values, the more stable the emulsion [31]. Therefore, when the concentration of HA is 0.175%, the change of Δ*BS* value is the smallest, suggesting the best stability of HA-β-emulsion. At the same time, the stability of Δ*BS* decreases first and then increases with the concentration of HA; thus, the results of MPS and rheological properties are consistent. As the concentration of HA increases in a range (under 0.2% *w*/*v*), the emulsion becomes thicker. It leads to fewer collisions and droplet mobility for the coalescence and sedimentation reduction [32,33]. Chevalier, Gomes, and Cunha [34] also found that the stability of the emulsion increased with increasing viscosity. The Δ*BS* values of the 0.175% HA-β-carotene emulsion are the smallest, showing the stability of the emulsion system. According to the above results, 0.175% of HA-β-carotene emulsion also has the lowest MPS and the highest EY. Overall, the emulsion is most stable when the HA concentration is 0.175%. 

The thermal stability of HA-β-carotene emulsion is shown in Figure 4A. With the increase in temperature and the prolongation of heat-treatment time, the MPS increases from 4.0 ± 0.2 µm to 4.7 ± 0.1 µm. However, the change of MPS mainly occurred in the first 20 min. At the same time, the effect of the three heat-treatment conditions on the size of the emulsion is less than 25%, which indicates that the treatment temperatures have little influence on the size of the emulsion. After being heated at 35 °C for 60 min, the emulsion’s MPS changed less than 0.1 μm and slightly increased with the elevation of heat-treatment temperature. After being treated at 60 °C for 20 min, the emulsion’s MPS increased about 0.3 μm. However, with the extension of heat treatment time, the MPS tended to be balanced. This may be due to the formation of a dense film on the surface of the β-carotene emulsion by HA and the polymerization of droplets within a short time (60 min), effectively preventing the continued increase of the MPS. Under the condition of 85 °C heat treatment, the HA-β-emulsion had a similar situation. The emulsion’s MPS only increased slowly after being heated for 20 min, and the thermal stability of the emulsion was good.

The effect of Na^+^ concentration on the MPS and ZP of HA-β-carotene emulsion with different HA concentrations is shown in Figure 4B. As the concentration of Na^+^ increases, the MPS of the emulsion has no significant change, which is similar to reported results [35,36]. Sherafati et al. [36] found that HA had no significant effect on the MPS of the carrot juice system. On the other hand, as the concentration increases, the ZP of the emulsion increases from −59.0 to −52.2 mV. This may be caused by the addition of Na^+^, which increased the cationic content in the emulsion. The stabilizing effect of HA on the emulsion system is mainly reflected in the spatial network structure formed by cooling [35]. 

The effect of pH values (between 2 to 6) on the MPS and ZP of HA-β-carotene emulsion with different HA concentrations are shown in Figure 4C. The MPS of emulsion increases from 4.0 to 5.9 μm with the decrease of pH value. This is because, with the decrease of pH, the degree of protonation of carboxyl groups gradually increases, which in turn reduces the electrostatic repulsion between droplets [7]. The absolute value of ZP decreased from 58.5 to 49.0 mV with the decrease in pH value. HA is a kind of anionic polysaccharide: its pKa value is about 3.525. When the pH value is less than 4, the negative charge density of gellan gum decreases. Overall, pH has little effect on the stability of HA-β-carotene emulsion systems in the pH range of 4 to 7.

### 3.4. The Changes of MPS, ZP, and Micromorphology of Emulsions during the In Vitro Simulated Digestion

The changes in the MPS and ZP of emulsions during the simulated digestion period are shown in Figure 5A. The MPS of the emulsion increases first and then decreases during digestion. After the simulated oral digestion phase (SODP) and simulated gastric digestion phase (SGDP), the MPS of all emulsions slightly increased from 4.5 ± 0.1 μm to 7.0 ± 0.1. Although salivary amylase added in SODP can hydrolyze α-1-4 glycosidic bond, it cannot destroy HA and cause a great deal of droplet aggregation due to its short-acting time. After 10 min of the simulated intestinal digestion phase (SIDP), the MPS of the emulsions increased sharply from 7.0 ± 0.1 μm to 20.0 ± 0.1 μm (*p* < 0.05). This might be because of the hydrolysis of the α-1-4 glycosidic bond of HA by pancreatic amylase in the small intestine. Further, the low concentration of HA may be affected by bile salts in the small intestine. We found a similar pattern in low concentrations of pectin, where pectin showed a higher tendency to bind to bile salts, favoring β-carotene micellization [37]. Finally, after SIDP, the MPS value of the emulsion dropped significantly to 1.0 ± 0.1 μm (*p* < 0.05). This result is caused by the destruction of the HA structure on the surface of the oil droplets. The ZP value changes slightly after SODP and SGDP from −58.0 ± 0.1 mV to −56 ± 0.1 mV. After 10 min of SIDP, the ZP value rises to −51.0 ± 0.1 mV. After SIDP, the ZP value rises sharply to −34 ± 0.1 mV. The increase in ZP value supports the process of gradual destruction of the HA structure.

The change of fat in HA-β-carotene emulsion after SIDP is shown in Figure 5B. The fluorescence density of SODP is not significantly reduced; that is, the oil does not have a notable reaction. After SODP, the accumulation of fat occurs at a low concentration of 0.100% HA, and it can be inferred that the fat in the high-concentration emulsion is not digested. After SGDP, the fluorescence density drops slightly. This may be because a small part of the emulsion is destroyed by the simulated gastric juice and aggregated to form a small number of oil droplets with a larger particle size. This is consistent with the MPS test results. Similar to Liu et al. [38], the sesame oil-based high-purity diacylglycerol oil-in-water emulsions they prepared appeared to accumulate in larger droplets after the SIDP stage. After the SIDP, the final digestion juice is almost no fluorescence; that is, there is almost no oil in it. This is because, after SIDP, pancreatic lipase has almost completely hydrolyzed oil [39].

### 3.5. Effect of HA as Emulsifier on the Release of FFA 

The relationship between the FFA release rate of HA-β-carotene emulsion and time during SIDP is shown in Figure 6A. The sustained release of biomolecules in structured materials development for specific applications has an important relationship with their biological activities [38,40]. After the emulsion entered the small intestine, the triglyceride was quickly broken down into diglyceride or glycerol monoesters and FFA by lipase. In the SIDP, the volume of NaOH used to maintain the neutral pH of the system can be converted into the release rate of FFA, which is an important index to measure the active components of bioaccessibility. FFA release rate varies greatly in the first 30 min of SIDP. As the concentration of HA increases from 0.100% to 0.175%, FFA release rate rises sharply, reaching a maximum of 70 ± 3% at a concentration of 0.175% and decreasing to 60 ± 1% at a concentration of 0.200%. The FFA release rate of the control group rises relatively slowly from 5 ± 1% to 22 ± 1%. This shows that most of the fat is digested at this stage. Later, the increase in the release rate slows down, with an average increase of approximately 10 ± 1%, while the control group rises to 25 ± 1%, which is slower than the experimental group. After 120 min of the SIDP, the final FFA release of emulsions with different HA concentrations is relatively close, and the release rate of FFA in emulsion with 0.175% HA concentration was 89.75%, which is significantly (*p* < 0.001) much higher than that in the control group (31.5%). Liu et al. [38] found that the FFA release rate increased swiftly in the first 10 min and finally reached over 80% after 120 min simulated digestion. The results showed that the oil digestibility could be improved effectively by using an emulsion as a carrier system. This is because the surface area of the droplet increases when water and oil are mixed to form a small droplet, which increases the rate of hydrolysis of the fat, which is consistent with Joung et al. [40].

### 3.6. Analysis of β-Carotene Bioaccessibility in HA-β-Carotene Emulsion

The effect of HA concentration on the bioavailability (%) after in vitro digestion (B) is shown in Figure 6B. The β-carotene bioaccessibility of the HA emulsion entrapped with MCT was significantly (*p* > 0.001) higher than that of the soluble β-carotene control group (4.1 ± 0.0%). The bioaccessibility of the emulsion first increases and then decreases as the HA concentration increases. When the HA concentration is 0.175%, the bioaccessibility rate is 26.6 ± 1.3%, which is the highest. When the HA concentration is 0.100%, 0.150%, and 0.200%, the bioaccessibility rate is 18.4 ± 1.3%, 22.2 ± 1.4%, and 25.0 ± 1.2%, respectively. It is believed that the more complete the lipid digestion, the more fat-soluble β-carotene encapsulated in the oil particles are released. During the process of fat hydrolysis in the SIDP, the nutrients carried by the emulsion are gradually released from the oil droplets. Composed of bile salts, phospholipids and FFA are absorbed by small intestinal epithelial cells in the form of micelles [41]. As gellan gum is resistant to enzymatic hydrolysis and with high negative charges, it may inhibit the adsorption of negatively charged lipases (and other anionic substances such as bile salts) to the droplet surface [42]. Therefore, the formation of lipid digestion products and mixed micelles can be inhibited to a greater extent [43,44]. In this way, the digestion rate of carotene is slowed down to achieve a good slow-release effect. Overall, there was a positive correlation between the degree of fat hydrolysis and the bioaccessibility of β-carotene.

## 4. Conclusions

This study applied HA as a natural emulsifying agent and evaluated its potential as an emulsifier and stabilizer in β-carotene emulsions. The effects of HA on the emulsifying effect and bioaccessibility of β-carotene emulsions during in vitro digestion were studied. Experiments showed that HA has a good emulsifying ability for β-carotene emulsion. HA showed good slow-release ability and enhanced β-carotene absorption in simulated digestion. This will widen the application of HA in the food industry and provide theoretical reference and basis for the preparation of emulsion systems with macromolecular polysaccharide emulsifiers as well as the study of digestibility. 

## Figures and Tables

**Figure 1 foods-11-01742-f001:**
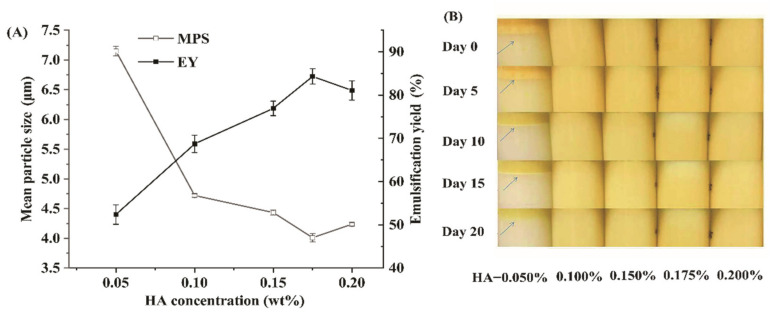
HA concentration’s effect on the MPS and EY (**A**) and storage stability (**B**) of HA-stabilized β-carotene emulsion during 20 days of storage.

**Figure 2 foods-11-01742-f002:**
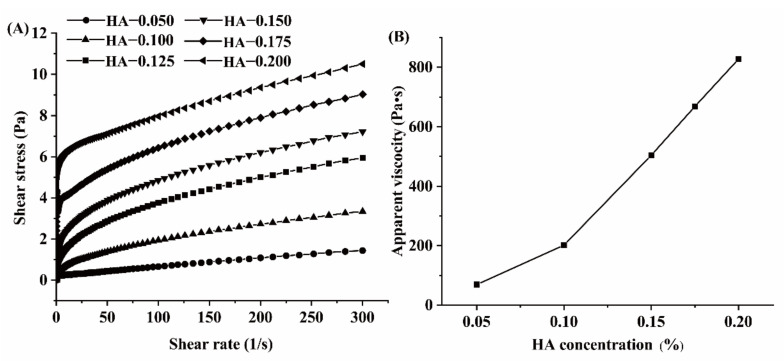
Relationship between shear stress and a shear rate of different HA concentrations of HA-β-carotene emulsion (**A**); effect of different HA concentrations of HA-β-carotene emulsion on mean apparent viscosity when the shear rate is 100 s^−1^ (**B**).

**Figure 3 foods-11-01742-f003:**
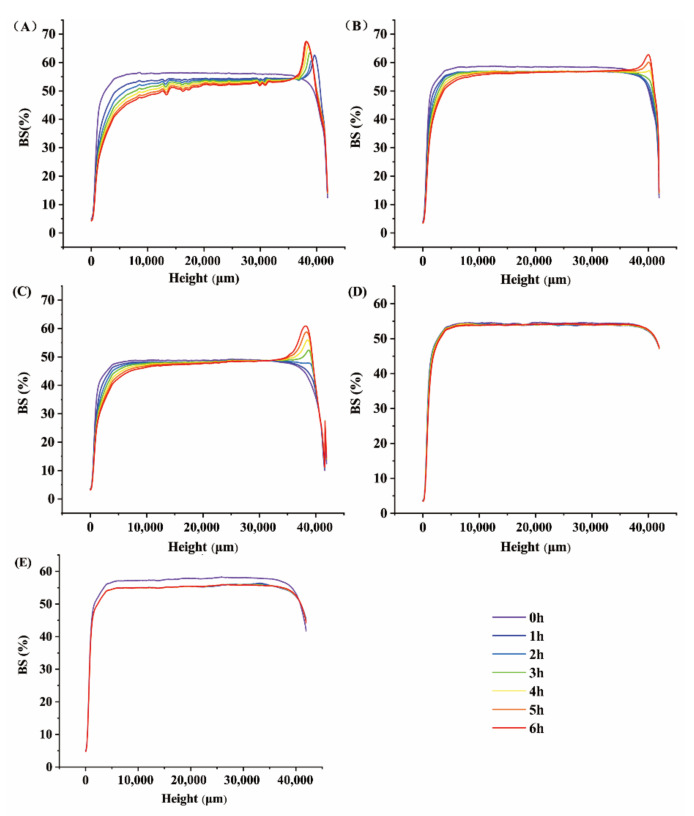
Effect of HA concentration of 0.100% (**A**), 0.125% (**B**), 0.150% (**C**), 0.175% (**D**), and 0.200% (**E**) on the stability of β-carotene emulsion.

**Figure 4 foods-11-01742-f004:**
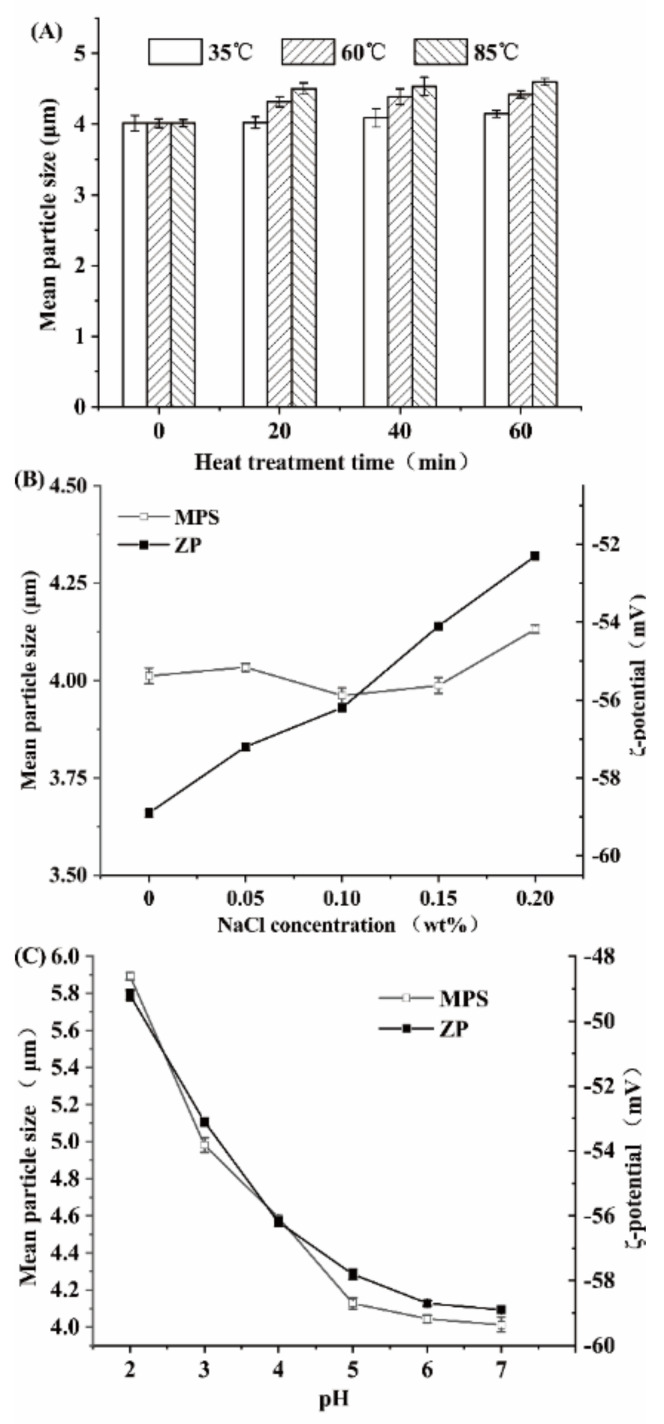
Effect of temperature (**A**), NaCl salt (**B**), and pH (**C**) on the stability of HA-stabilized β-carotene emulsion.

**Figure 5 foods-11-01742-f005:**
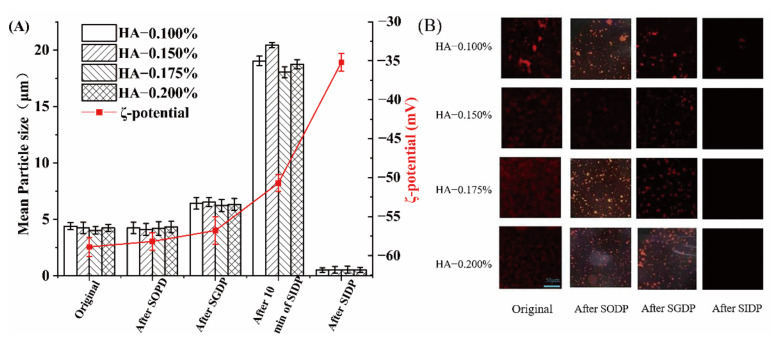
The changes of the MPS and ZP of emulsions during the simulated digestion period (**A**); laser confocal microscopy of HA-β-carotene emulsion in the SIDP (**B**).

**Figure 6 foods-11-01742-f006:**
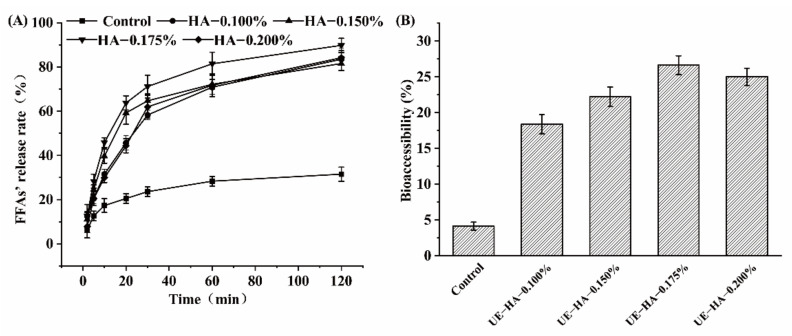
The amount of FFA released from the HA-β-carotene emulsion and the control group in the SIDP (**A**); influence of different emulsification methods on the bioaccessibility (%) of in vitro digestion (**B**).

## Data Availability

The data presented in this study are available on request from the corresponding author.

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
