# Peer review of "Fabrication of High-Acyl Gellan-Gum-Stabilized β-Carotene Emulsion: Physicochemical Properties and In Vitro Digestion Simulation"

_foods, 2022, doi:10.3390/foods11121742_

Round 1

Reviewer 1 Report

Review of MS titled: “Fabrication of high acyl gellan gum stabilized β-carotene emulsion:

      physicochemical properties and in vitro digestion simulation”

By: Yuecheng Meng et al.

General comments

This work has been soundly conducted. The use of the scientific method has been used, and the novelty relies on using a known hydrocolloid as emulsifier and the evidence in vitro, that it can be used for significantly better release of beta-carotene during simulated digestion.

However, the reference list shows about 33% of publications from the year 2017 or newer. Therefore, the percentage of these references should be increased. There are also some minor points that require correction.

Summary:

  1. L 12 The authors should state the oil phase and its proportion, used to produce the emulsion system.
  2. L. 15, 52, 54, 357. The words “in vitro” should be italicized.
  3. L. 17. The authors claim that 3.0 mV of ξ-potential provides good stability, when it is well known that good stability is achieved at absolute values of 30 mV or higher. Please clarify.

Specific  comments:

Title: the words “in vitro” should be italicized

L. 34-35. The microorganism genus and species should be in italics.

L. 59. Can the authors give more details about the acyl content of gellan gum, mean molecular weight, and its rheological properties? Please clarify.

L. 65. Can the authors explain the meaning of “so on”?

L. 72. Can the authors describe more precisely the composition of the medium-chain triglyceride? Please clarify.

L. 73. The β-carotene concentration was not varied. Can the authors explain why? Please clarify.

L. 76. The superscript of s-1 should be corrected. Please correct.

L. 91. The equation (1) is not clear in relation to the subtraction of unwrapped β-carotene from 1, while dividing the result by the total β-carotene. Apparently, the subtraction should be from the total β-carotene, not from 1. Unwrapped β-carotene may be defined for better understanding. Please clarify.

L. 120. What was the HA concentration, and pH of the emulsion used here?

L. 145-147. It is not clear what phase represents the concentration of β-carotene in micelles solution, and what phase represents β-carotene concentration in the digestive tract after digestion. Please clarify.

L. 164. According to Fig. 1, the EY is about 84% at 0.175% HA. The value shown here should be explained. Please clarify.

L. 168. According to Fig.1, the emulsions containing 0.175% HA show a slight creaming after 10 days of storage. Can the authors explain this effect?

L. 172. It is not clear if the emulsifying effect of HA is due to the high acyl group content (high negative charge, L. 178-179) or hydrophobic interactions with the oil phase. Please clarify.

L. 182-183. The authors mention that HA can coat the oil droplets by adsorption on the interface of the oil drop. However, they do not provide evidence for this statement. Please clarify.

L. 196. From Figure 2, it sems that at the last three concentrations of HA, there is yield stress before fluid flow. Can the authors clarify this behavior?

Figure 2. the label of Fig. 2A, showing HA concentrations, shows one of 1.250%, which I believe is not correct. Please clarify.

L. 212. Can the authors give the meaning of BS? Please clarify.

L. 226. I consider that the term HA- β- term should include the word carotene after the beta term. In addition, the values of MPS should be disclosed. Please clarify.

L. 259-260. It is not clear why the electrostatic repulsion decreases at pH 7, since the negative electric charge is at its highest, giving the highest zeta potential. Please clarify.

L. 262-263. Since the pKa of HA is 3.525, below this pH the particles are mainly protonated and thus the negative charge density decreases, not increase, as written here. Please clarify.

L. 340. It is not clear why the bioaccesibility corresponds to the results of FFA’s release rate. Please clarify the methodology used to measure this term.

L. 358. The authors should explain why the HA emulsion of beta-carotene has a good slow-release ability of this compound.

Author Response

Dear reviewer,

We would like to thank you for the careful reading of our manuscript. Your comments are valuable and very helpful for revising and improving our paper. We have revised the manuscript based on your comments. Please see the attached file for details.

Reviewer 2 Report

This study applied high acyl gellan as a natural emulsifying agent and evaluate its potential as an emulsifier and stabilizer in β-carotene emulsions. It is organized well and needs some minor corrections as follows:

Line 40: there is an interest….

Line 46: how the authors stated that there is rarely wok on the emulsification of high acyl gellan. Please read the following references, and more literature review is required for this work.

High acyl gellan as an emulsion stabilizer

Emulsions stabilized by high acyl gellan and KCl

Rheological analysis of emulsion-filled gels based on high acyl gellan gum

Interfacial rheology of sodium caseinate/high acyl gellan gum complexes: Stabilizing oil-in-water emulsions

Line 96: use superscript for s-1.

Author Response

(The authors gave the same response as above.)
